# Steady-State PERG Adaptation Reveals Temporal Abnormalities of Retinal Ganglion Cells in Treated Ocular Hypertension and Glaucoma

**DOI:** 10.3390/diagnostics15141797

**Published:** 2025-07-16

**Authors:** Tommaso Salgarello, Andrea Giudiceandrea, Grazia Maria Cozzupoli, Martina Cocuzza, Romolo Fedeli, Donato Errico, Antonello Fadda, Filippo Amore, Marco Sulfaro, Epifanio Giudiceandrea, Matteo Salgarello, Stanislao Rizzo, Benedetto Falsini

**Affiliations:** 1Eye Clinic, Fondazione Policlinico Universitario A. Gemelli IRCCS, Università Cattolica del Sacro Cuore, Largo A. Gemelli 8, 00168 Rome, Italy; andrea.giudiceandrea@policlinicogemelli.it (A.G.); stanislao.rizzo@policlinicogemelli.it (S.R.); bfalsini@gmail.com (B.F.); 2Institute of Ophthalmology, Università Cattolica del Sacro Cuore, 00168 Rome, Italy; 3Institute of Ophthalmology, Pia Fondazione Cardinale Giovanni Panico Hospital, 73039 Tricase, Italy; mgcozzupoli@gmail.com (G.M.C.);; 4Department of Cardiovascular, Dysmetabolic and Aging Associated Diseases, Istituto Superiore di Sanità, 00161 Rome, Italy; antonello.fadda@iss.it; 5National Centre of Services and Research for the Prevention of Blindness and Rehabilitation of Low Vision Patients, Fondazione Policlinico Universitario A. Gemelli IRCCS, 00168 Rome, Italy; f.amore@iapb.it (F.A.);; 6San Paolo Hospital, University of Milan, 20122 Milan, Italy; 7Faculty of Medicine and Surgery, Università Cattolica del Sacro Cuore, 00168 Rome, Italy; matsalga16@gmail.com

**Keywords:** glaucoma, pattern electroretinogram, optic nerve, adaptation

## Abstract

**Background/Objectives:** This study investigates adaptive changes in long-lasting pattern electroretinogram (PERG) responses in ocular hypertension (OHT) and open-angle glaucoma (OAG) patients, and in healthy subjects. **Methods:** Sixty consecutive individuals were recruited, including 20 OHT, 20 OAG, and 20 normal subjects. All participants underwent comprehensive ophthalmologic examination, 30–2 perimetry, and retinal nerve fiber layer imaging. Steady-state (7.5 Hz) PERGs were recorded over approximately 2 min, in response to 90% contrast alternating gratings within a large field size. The recordings were acquired into a sequence of 10 averages (packets), lasting 10 s each, following a standardized adaptation paradigm (Next Generation PERG, PERGx). Key outcome measures included PERGx parameters reflecting response amplitude and phase changes over time. **Results:** The PERGx grand average scalar amplitude, a surrogate of ordinary PERG, was significantly reduced in both OHT and OAG groups compared to normal subjects (*p* < 0.01). In contrast, minimal adaptation changes were noted in PERGx amplitude among all groups. The PERGx phase exhibited a progressive decline over time, with consistent delays of approximately 20 degrees across all groups. Angular dispersion of the PERGx phase increased significantly in OHT patients compared to normal subjects (*p* < 0.05). An inverse relationship was observed between PERGx angular dispersion and treated intraocular pressure, specifically in OHT patients. **Conclusions:** The findings suggest that both OHT and OAG eyes may exhibit temporal abnormalities in PERG adaptation, potentially indicating early dysfunction in retinal ganglion cell activity. **Translational Relevance:** PERGx phase changes may have significant implications for glaucoma early detection and management.

## 1. Introduction

The pattern electroretinogram (PERG) is a validated electrophysiological test widely employed to assess retinal ganglion cells (RGCs) function in optic neuropathies, particularly in glaucoma [1]. Alterations in the PERG signal can precede structural damage and overt RGC loss [2] and are, therefore, considered potential early indicators of dysfunction, especially in patients with ocular hypertension.

To better understand disease progression, mathematical modeling approaches have been applied to PERG data, simulating the transition from physiological to pathological states using adaptable parameters. These models support hypothesis testing and curve fitting strategies [3]. Furthermore, assessing PERG responses to sustained visual stimuli has provided insights into RGC vulnerability under stress and into transient dysfunctions that may be reversible with pharmacological intervention [3].

“Adaptation” describes a physiological/metabolic response of RGCs when exposed to energetically stressful conditions. A prior study involving 28 healthy eyes showed that continuous stimulation with high-contrast, rapidly reversing patterns leads to a progressive reduction in PERG amplitude, following an exponential decay until a plateau is reached [4]. The amplitude of the PERG response decreased progressively, typically stabilizing at around 30% of the baseline amplitude after approximately 110 s of stimulation.

This adaptive phenomenon likely reflects the mismatch between the metabolic demands of RGCs and the energy supply available during prolonged exposure to high-frequency, steady-state (SS) visual stimulation (typically lasting 100 s or more) [5]. In response, neurons adjust their activity to maintain a balance between functional output and energy resources. In healthy eyes, this results in a progressive decline in PERG amplitude over time, a feature commonly described as “adaptation”. A reduced capacity for adaptation, potentially linked to impaired vascular autoregulation or insufficient neural modulators, may indicate early dysfunction of retinal ganglion cells [5].

Porciatti et al. [6] were among the first to investigate SS-PERG adaptation in glaucomatous eyes, comparing patients with early-stage primary open-angle glaucoma (OAG), glaucoma suspects (GSs), and normal subjects. Their findings indicated that the magnitude of adaptive PERG phase changes significantly decreased with increasing severity of disease, whereas amplitude adaptation showed no significant variation among the groups.

Our group recently applied the novel optimized Next Generation PERG (PERGx as a contraction of PERGnext) protocol in glaucoma patients (pre-perimetric and early to moderate taken together) compared to normal [7]. The results demonstrated that glaucomatous eyes exhibited a flatter, less negative PERGx amplitude slope compared to healthy eyes, pointing to compromised RGC adaptability in these patients.

Building upon those findings, the present study aims to investigate whether PERGx adaptation abnormalities are also present in eyes with ocular hypertension (OHT). Specifically, we sought to compare PERGx parameters—including amplitude slope and phase variability—among eyes with OHT, OAG, and normal subjects, to determine whether early functional impairments can be detected in the absence of overt glaucomatous damage.

## 2. Methods

A cross-sectional investigation was conducted on consecutive patients referred to the Glaucoma Service of Fondazione Policlinico Universitario A. Gemelli IRCCS–Università Cattolica del Sacro Cuore in Rome, Italy. Data collection took place between November 2021 and March 2022. The study protocol (ID 3701) was approved by the local Ethics Committee, and all procedures complied with the principles outlined in the Declaration of Helsinki. Written informed consent was obtained from all participants following a detailed explanation of this study’s objectives and procedures.

**Subjects.** This study enrolled 60 individuals, consisting of 20 normal subjects (NSs), 20 OHT, and 20 primary OAG patients (58.3% women and 41.7% men; mean age ± standard deviation [SD]: 59.8 ± 10.1 years, range 40–80). Only one eye per participant was included in the analysis. For patients with bilateral eligibility, the eye with more reliable test performance or more complete data was selected.

Age and sex distribution were comparable across all study groups. All participants—both patients and normal subjects—underwent a complete ophthalmologic evaluation, including best-corrected visual acuity (BCVA) using a Snellen chart, slit-lamp examination of the anterior segment and fundus, and intraocular pressure measurement with Goldmann applanation tonometry. Within one week, additional tests included central corneal thickness assessment via ultrasonic pachymeter (Pachmate DGH55, DGH Technology, Inc., Exton, PA, USA), standard automated perimetry using the 30-2 white-on-white protocol (Humphrey Field Analyzer 750i, Carl Zeiss Meditec, Inc., Dublin, CA, USA), SS-PERG recording with an adaptation protocol (Retimax, CSO, Florence, Italy), and measurement of both peripapillary retinal nerve fiber layer (RNFL) and macular ganglion cell/inner plexiform layer (GCIPL) thicknesses by spectral domain Cirrus HD-OCT (model 5000, sw. version 10.0, Carl Zeiss Meditech, Inc., Dublin, CA, USA).

Following the criteria previously adopted by our group, subjects were enrolled based on the following inclusion parameters: age between 40 and 80 years, normal range central corneal thickness values (520–580 µm), and fulfillment of diagnostic requirements for OAG. These included (1) documented IOP >21 mmHg on at least two separate visits; (2) optic nerve head abnormalities on slit-lamp examination with a 78-diopter lens, such as vertical cup/disc (C/D) ratio >0.6 in medium-sized optic discs (or corrected according to the physiological relationship between C/D ratio and disc size) [8], and/or inter-eye asymmetry in C/D ratio ≥0.2 not attributable to anatomical disc differences, focal or diffuse rim thinning, or notching; (3) reliable and reproducible visual field defects consistent with glaucoma based on the Hodapp–Parrish–Anderson classification [9]. To qualify for the OAG group, all three criteria had to be met. In contrast, OHT patients only needed to meet the elevated IOP criterion in the absence of optic disc or visual field damage.

All patients with OAG or OHT were receiving topical ocular hypotensive treatments, such as β-blockers, prostaglandin analogues, carbonic anhydrase inhibitors, or α2-agonists, achieving a stable IOP below 21 mmHg. Some patients were also taking neuroprotective agents like citicoline.

Exclusion criteria included BCVA worse than 20/25, spherical equivalent >±2.00 D or astigmatism >1.00 D, optic disc pallor exceeding cupping, previous cataract surgery, recent changes (within 3 months) in IOP-lowering and/or neuroprotective therapy, unreliable visual fields, and any ophthalmic or neurological condition potentially affecting visual function or test performance.

**Perimetry.** Standard automated perimetry (SAP) was performed using the Humphrey Field Analyzer with the 30-2 SITA-Standard strategy. Only reliable visual field tests, defined by fixation losses and false positives or negatives <20% [10], were included in the analysis. Abnormal perimetry was defined as a typical reproducible defect (arcuate and/or paracentral scotoma or nasal step) in three consecutive exams [11], with one or more of the following alterations: Glaucoma Hemifield Test outside normal limits, pattern standard deviation (PSD) with *p* <5%, and a cluster of ≥3 adjacent points, not contiguous with the field borders nor the blind spot, in the upper and/or lower hemifield of the total and pattern deviation plots with *p* <5%, one of which reached *p* <1%. For data analysis, the two global indices of field sensitivity, MD and PSD, were collected.

**OCT.** Spectral-domain OCT imaging was carried out using the Cirrus HD-OCT system to assess both peripapillary RNFL and macular GCIPL. Scans were performed using the Optic Disc Cube 200 × 200 protocol for RNFL evaluation and the Macular Cube 512 × 128 protocol for GCIPL analysis. The OCT lens was calibrated according to the individual refractive error. From these acquisitions, average thickness values of the RNFL and GCIPL were extracted for further analysis.

**PERGx recording and analysis.** PERG signals were recorded from one eye at a time, using 10-mm adhesive skin electrodes positioned beneath the lower eyelid. The non-tested eye served as a reference, while the ground electrode was placed on the forehead. Subjects maintained fixation on a central target within a 60° × 50° visual field presented at a distance of 57 cm. Testing was conducted under natural pupil conditions and with full refractive correction. Signals were amplified (100 K), filtered (1–250 Hz bandwidth, 6 dB/octave slope), digitized (12-bit resolution, 2 kHz sampling rate, 100 µV AC range), and averaged in synchronism with stimulus onset. Artifacts, mainly from blinks or large eye movements, were automatically rejected to minimize amplitude bias.

**PERG adaptation protocol.** PERGx was recorded according to a published protocol [7], using a commercially available instrument (Retimax CSO, Florence, Italy). Stimuli were black-and-white bars displayed as horizontal gratings (0.8 cycles/degree, 95% Michelson contrast, 35 cd/m^2^ mean luminance), alternating in counterphase at 7.5 Hz (15 reversals/s). They were delivered continuously over a 2 min period on a high-resolution OLED screen. The response was recorded as a sequence of 10 partial averages (packets), each lasting 10 s and comprising 60 stimulus cycles. The first packet, obtained in response to a grey background matched in luminance to the test pattern, was used to estimate baseline noise. Two replications of the entire adaptation paradigm, including noise level assessment, were recorded, and an appropriate time interval between replications was chosen to avoid residual adaptive effects.

A frequency-domain (Fourier) analysis was conducted on the recordings to isolate the PERG second harmonic (2P), which represents the principal component of the SS-PERG experiments [12].

The resulting nine waveforms per patient (excluding the initial noise waveform) were analyzed by plotting the amplitude and phase of the second harmonic (2P) against time for each individual. At the conclusion of this process, each subject’s response comprised nine second harmonic packets. Furthermore, PERGx amplitude and phase values were averaged across group subjects both for single packets (average scalar amplitude and phase) as well as for each response over the nine packets (grand-average scalar amplitude and phase). We assumed that the grand-average scalar parameters represented an index of non-adapted RGC activity and corresponded to the ordinary SS-PERG.

In addition to these averages, we studied the PERGx adaptation in terms of the delta of scalar amplitude (estimated from the linear regression, see below) and the angular dispersion of the response phase (i.e., phase circular SD). Regarding the delta amplitude, it represents the amplitude difference throughout the PERGx procedure derived from the amplitude slope of the linear regression, automatically applied to the PERGx amplitudes as a function of packets’ order number. The residuals of this regression provide an estimate of the “noise”, that is, the component of variance not attributable to the adaptive modifications. Phase angular dispersion is considered an index of PERGx phase variability during the adaptation process.

**Statistical analysis.** Forty-five right eyes and 15 left eyes -one randomly selected per subject or patient- were included in this study. No interocular asymmetries were observed in any patient, ensuring that both eyes met the criteria for OAG or OHT. The primary outcome measures selected for this study were the following standard PERGx parameters: average scalar amplitude and phase per packet; delta amplitude and phase angular dispersion, representing adaptive PERGx changes; and grand-average scalar amplitude and phase, as surrogates of ordinary non-adapted SS-PERG. The sample size study was estimated based on the previous studies conducted by our group.

Statistical analyses were performed using SPSS software version 15.0 (IBM SPSS, Armonk, NY, USA) and Origin software version 6.0 (Microcal Origin, Microcal Software Inc., Northampton, MA, USA). Alpha and beta errors were set at 5% and 20%, respectively. The nine electrophysiological signal packets were separately analyzed, and PERGx amplitude and phase were studied as individual temporal series, as well as averaged across group subjects and plotted as a function of sequential packets. Linear regression analyses were then applied to the amplitude and phase data to evaluate the presence of adaptive behavior in each study population. The following variables were considered as continuous quantitative variables: age; IOP measurement; perimetric MD and PSD indices; OCT RNFL and GCIPL thicknesses; PERGx grand-average scalar amplitude and phase, delta amplitude, and phase angular dispersion. Assimilability to normal distribution was verified using the Kolmogorov–Smirnov test.

The one-way analysis of variance (ANOVA) was used to determine whether there were any statistically significant differences in the demographic, morphological, and functional parameter means among the three independent groups (glaucoma patients, OHT patients, and normal subjects). The post hoc Tukey test was also performed to determine individual group differences.

Additional correlational analyses were conducted to investigate the association among electrophysiological parameters and between them and treated IOP across groups. Specifically, PERGx phase angular dispersion values were plotted against (1) delta amplitude values, (2) grand-average amplitude values, and (3) IOP values within the OHT group.

## 3. Results

Descriptive statistics (demographic and clinical data: age, gender, IOP; perimetric data: MD, PSD) for the study groups are summarized in Table 1. Statistically significant differences were observed between the groups in terms of age and treated IOP at enrollment. Specifically, the mean treated IOP was significantly higher (*p* < 0.01) in OHT eyes compared to OAG and normal eyes. No significant differences were found in age (*p* = 0.31), indicating that the groups were comparable in terms of age distribution. Likewise, gender distribution did not differ significantly across the groups.

Table 2 summarizes the primary and secondary outcomes for the study groups, including PERG parameters (delta amplitude, phase angular dispersion, grand-average amplitude, and phase) and OCT parameters (average RNFL and GCIPL thicknesses).

Phase angular dispersion also differed significantly across the groups (*p* < 0.05), with higher mean values in both patient groups compared to normal subjects. A significant difference was observed between OHT patients and normal subjects (*p* < 0.05), whereas no significant differences were found between OAG and OHT or normal subjects.

The grand-average scalar amplitude was also significantly different (*p* < 0.01) among the groups; normal subjects exhibited a significantly greater mean scalar amplitude compared to both OHT (*p* = 0.01) and OAG patients (*p* < 0.01), whereas values from OHT patients were not statistically different from those from OAG patients. Additionally, the mean grand-average scalar phase was significantly different (*p* < 0.01) among the groups, being more advanced in OHT and OAG patients compared to normal subjects (*p* < 0.05), and without significant differences between OHT and OAG patients. 

For the secondary outcomes, average RNFL and GCIPL thicknesses were significantly lower in the OAG group compared to both OHT patients and normal subjects (*p* < 0.01), but not statistically different between OHT patients and normal subjects.

Figure 1A,B illustrates the time course of both PERGx amplitude and phase from the first to the ninth packet, after excluding initial noise data packets, across the three study groups. In Figure 1A, the plots show greater PERGx amplitude values (*p* < 0.01) as a function of packets in the normal subjects compared to the OHT and the OAG patients (see also the mean grand-average scalar amplitude in Table 2, regardless of the packet number). Normal and OAG eyes displayed a decline in mean amplitude over time, whereas no notable trends emerged in PERGx amplitude in OHT eyes.

In Figure 1B, the PERGx phase was advanced in both OHT and OAG patients compared to normal subjects throughout the packets (see also the mean grand-average scalar phase in Table 2, regardless of the packet number), with a similar negative slope across all groups indicating a phase delay during sustained stimulation (i.e., adaptive changes).

Figure 2 displays PERGx phase angular dispersion values plotted against the corresponding delta amplitude values for the three study groups. Notably, 40% of OHT eyes exhibited a substantial (>45 degrees) increase in phase dispersion compared to either normal subjects or OAG eyes. Furthermore, more than 60% of the OHT eyes with phase angular dispersion greater than 45 degrees demonstrated a positive delta in PERGx amplitude, indicating abnormal adaptation.

When analyzing the correlations between electrophysiological and clinical variables within each patient group, no significant associations were found between PERGx parameters and age or structural measures (RNFL and GCIPL thicknesses) in either the OHT or OAG groups. However, in the OHT group, a significant inverse correlation was observed between PERGx phase angular dispersion and IOP measured at the time of testing (r = −0.63, *p* = 0.003), as illustrated in Figure 3, indicating that higher IOP values were associated with reduced phase variability. This relationship was not observed in the OAG or normal groups, suggesting specificity to OHT.

Moreover, as shown in Figure 4, a strong negative correlation was found between PERGx phase angular dispersion and the grand-average PERGx amplitude (r = −0.68, *p* < 0.01) across both OHT and OAG eyes, with greater phase variability observed at lower signal amplitudes. Significant inverse correlations were also found between grand-average amplitude and angular dispersion in both OHT and OAG groups, with a particularly strong correlation between amplitude and delta in the OAG group (r = −0.72, *p* < 0.001).

No significant correlations between angular dispersion and IOP were found in the OAG and normal groups. The full set of correlation results is provided in Appendix A.

## 4. Discussion

The aim of this study was to assess and compare the adaptive changes in PERGx among three distinct groups: healthy subjects, ocular hypertensives, and glaucomatous patients.

This research builds on the previous research investigating PERGx adaptive changes in glaucoma patients (including both manifest and pre-perimetric glaucoma) in comparison to normal subjects.

Under stressful physiological conditions, such as sustained pattern stimulation, the healthy eye exhibits a decrease in PERGx amplitude and phase (i.e., delay) over time, indicating RGC adaptation [4]. In this study, PERGx adaptation during prolonged stimulation was more effectively captured by phase dynamics rather than by variations in amplitude, in line with prior evidence, highlighting the critical role of the PERGx phase in assessing RGC function and its adaptive capacity.

Interestingly, we observed not only a consistent trend towards phase delay as a function of packet/time in all three groups but also a significant increase in phase angular dispersion (an index of temporal variability), as indicated by greater SD, in OHT patients and, to a lesser extent, in OAG patients compared to the normal subjects. In this respect, over 60% of OHT eyes with high phase angular dispersion (specifically, exceeding 45 degrees) exhibited a positive delta in PERGx amplitude (indicative of reduced adaptation), pointing out that less adaptable amplitude responses are associated with greater variability in the timing of the PERGx signal. Thus, although phase shifts during PERGx recordings appeared to be more prominent than amplitude changes, a relationship between reduced amplitude adaptability and increased phase variability was evident.

The observed inverse correlation between IOP and phase angular dispersion was significant only in OHT eyes, suggesting a potential RGC dysfunction specific to ocular hypertension, despite IOP normalization through treatment. Conversely, in OAG eyes, although a similar trend was noted, the relationship was weaker and not statistically significant. These findings support the hypothesis that temporal PERGx alterations in OHT may reflect early and potentially reversible dysfunction.

The magnitude of phase adaptive changes was similar across all groups, suggesting that adaptation may occur even in both OHT and OAG patients, beyond healthy subjects. However, both OHT and OAG groups demonstrated greater phase dispersion compared to normal subjects, with a statistically significant difference observed specifically between OHT patients and normal subjects. Notably, these changes in phase dispersion may precede any detectable anatomical abnormalities in the inner retina or optic nerve fibers, highlighting their potential as a treatment-independent biomarker of early glaucoma-related dysfunction. Consequently, the sustained pattern stimulation seems to represent a sort of “stress test” for RGCs, wherein PERGx temporal abnormalities uncover subtle abnormalities not yet evident on anatomical imaging or perimetric tests [4].

Previous research by Porciatti reported that the magnitude of adaptive PERG phase changes significantly decreased with increasing severity of disease, whereas adaptive PERG amplitude changes were similar in the three groups of subjects (normal subjects, patients with suspicion of glaucoma, and patients with earliest stages of manifest open-angle glaucoma) [6]. The present study corroborates these earlier results and further elucidates a specific PERGx temporal abnormality consisting of increased phase dispersion. Additionally, an inverse correlation between PERGx phase dispersion and treated IOP at the time of testing was observed in the OHT group. Whether this variability is modulated by different hypotensive agents remains unclear. Neuroprotective strategies may also influence PERG dynamics, as previously proposed by Monsalve et al. [7]. In this respect, we excluded patients with recent changes in medical (hypotensive and neuroprotective) therapy to avoid confounding effects. It is tempting to speculate that the adaptive changes in PERGx phase may be influenced by the IOP regulation induced by different IOP-lowering medications. Notably, this inverse relationship suggests that phase variability persists despite IOP normalization and may even increase as IOP decreases. This pattern raises the possibility that certain hypotensive treatments, while effective in lowering IOP, could modulate RGC responsiveness or metabolic dynamics in a way that affects the temporal stability of the PERGx signal.

A more parsimonious, although not alternative, explanation might be that the RGC response shows increased phase variability due to a phase “jitter” effect, simply related to lower amplitude and signal-to-noise ratio [13]. Indeed, a lower amplitude response and adaptation may lead to increased phase variability, reflecting a diminished correlation between RGC firing and stimulus timing. In this regard, the strong inverse association observed between phase angular dispersion and grand-average PERGx amplitude (see Figure 4) suggests that PERGx abnormalities reflect an RGC dysfunction consisting of reduced firing and loss of temporal accuracy and predictability. From a clinical perspective, increased temporal variability during PERG adaptation may shed further light on early physiological changes in an otherwise anatomically preserved inner retina of OHT eyes. The increased PERG phase angular dispersion was already reported in glaucoma patients by Salgarello et al. (albeit not statistically different from normal subjects) and Mavilio et al. [14]. Our data confirm the results of both previous studies and extend their findings, suggesting that these changes may precede any detectable glaucomatous abnormality in OHT patients.

Findings by Colotto et al. [15] demonstrated early RGC dysfunction in OHT through PERG amplitude reduction during transient IOP elevation, along with consistent phase delays. However, their analysis focused on average phase shifts, without addressing intra-subject phase instability. Different from the study by Colotto, which identified early OHT dysfunction primarily via amplitude reduction, our work adds a novel dimension by assessing phase instability, thereby expanding the diagnostic potential of PERG in this population.

However, our study introduces two key advancements over previous works [6,16], which amplify the applicability of PERG habituation, in particular phase variability, as an early biomarker of dysfunction: (1) it examines even a population of pharmacologically treated OHT patients to determine whether phase variability emerges at an early stage and persists despite controlled IOP; (2) it employs the phase angular SD as a direct, instrument-derived measure of phase variability, defined as “Angular Dispersion”, offering a more objective and standardized measure than previously used CoV or LOWESS-based indices used in earlier studies [6].

Unlike Porciatti’s study, which used a modified PERGLA system and included mostly untreated glaucoma suspects and early glaucoma cases, we employed the standardized Next Generation PERG with built-in algorithms and evaluated a patient population of treated OHT and early-to-severe POAG eyes.

As for Monsalve et al. [16], they evaluated short-term PERG adaptation in a mixed population, including healthy controls, early manifest glaucoma patients under hypotensive treatment, and NAION patients, by using a different commercially available instrument validated during its development in a previous report of the same group (Monsalve et al., 2017) [11]. They concluded, in 2017, by suggesting that SS-PERG adaptation should be included in the clinical reports together with SS-PERG amplitude and phase/latency, with the amplitude adaptation accounting for a sizeable portion of the PERG signal, whereas the phase adaptation was small and not significant. However, their study included a limited number of glaucoma patients (*n* = 7), which may affect the statistical power and generalizability of their findings. In contrast, our study specifically explored adaptive changes in PERGx phase variability and their persistence despite effective treatment, utilizing a larger and different (OHT and POAG) patient cohort.

In conclusion, deepening the current notions in the literature regarding the relevant aspect of the adaptation of temporal dynamics, the present study suggests that phase dispersion, rather than amplitude reduction, may serve as a sensitive early marker of RGC dysfunction in ocular hypertension and glaucoma. Future studies should investigate whether phase dispersion is modifiable through neuroprotective strategies or specific classes of IOP-lowering drugs.

## Figures and Tables

**Figure 1 diagnostics-15-01797-f001:**
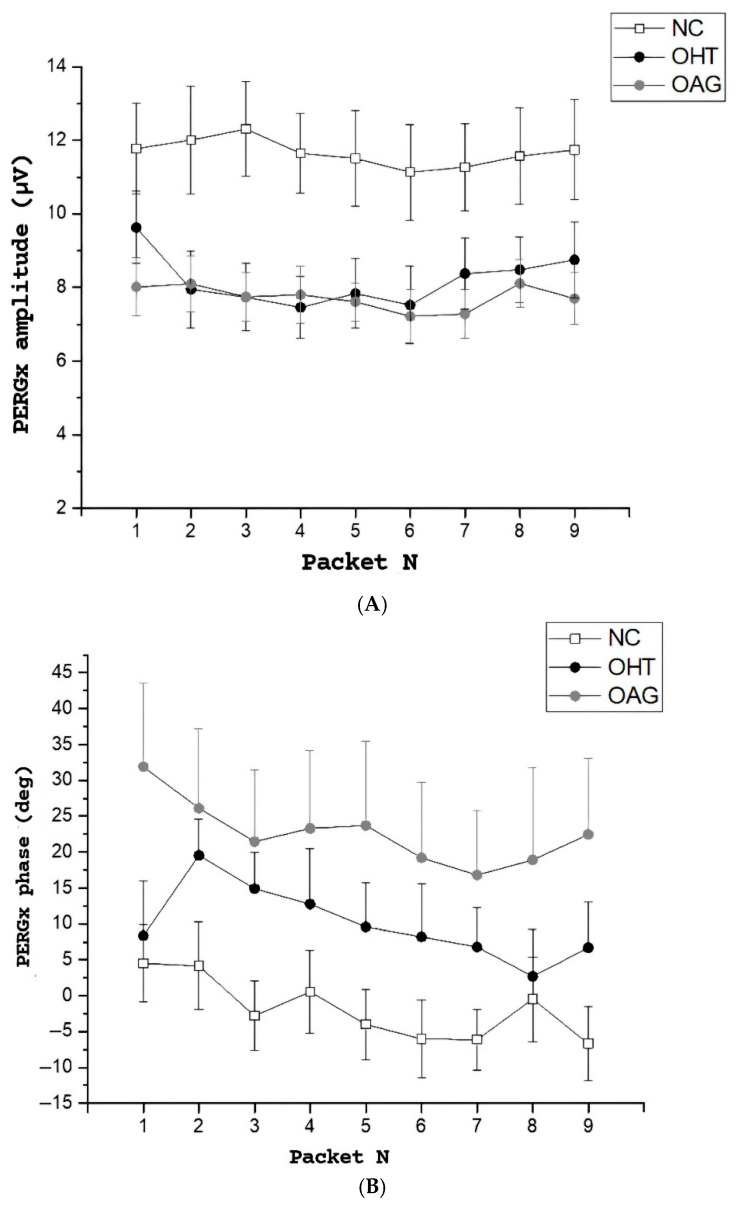
Scatter plots of PERGx scalar 2P amplitude (**A**) and phase (**B**) averaged across all participants from each study group (±SEM) as a function of packet number. The amplitude data show greater values in normal subjects (NC) with a slight decline over time, different from ocular hypertension (OHT) and open-angle glaucoma (OAG) patients. The phase was advanced in both OHT and OAG groups.

**Figure 2 diagnostics-15-01797-f002:**
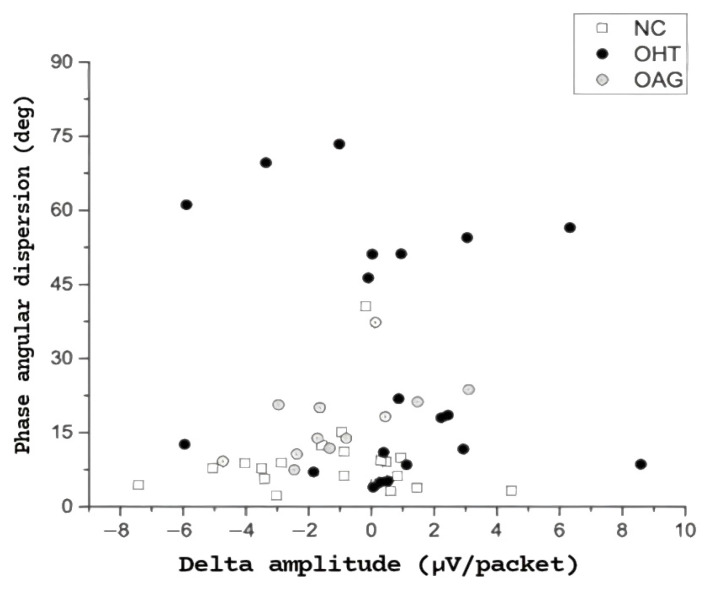
Individual PERGx phase angular dispersion values plotted as a function of corresponding PERGx delta amplitudes in the three study groups. A high proportion of OHT eyes showed a substantial increase in phase dispersion compared to the other groups’ eyes, whose values were greater than 45 degrees, demonstrating a mostly positive delta amplitude.

**Figure 3 diagnostics-15-01797-f003:**
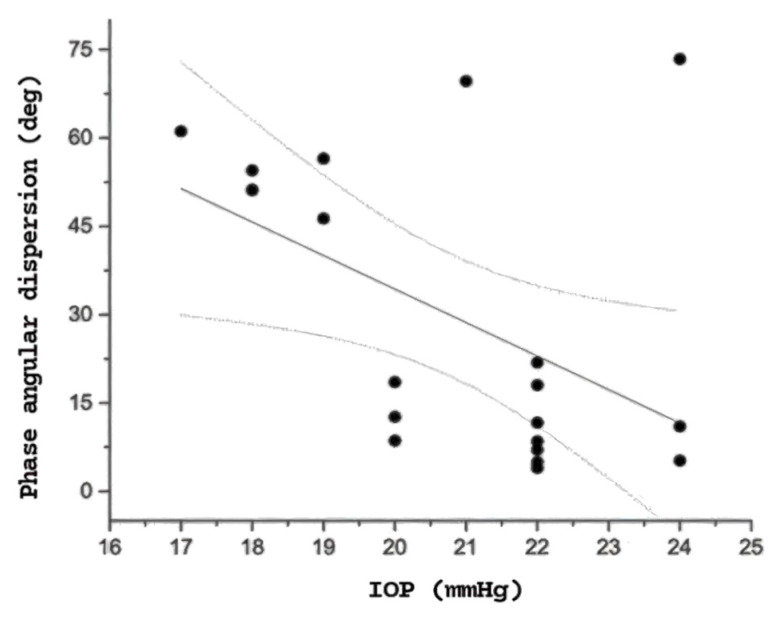
Relationship between PERGx phase angular dispersion values and treated IOP values in the OHT group at the time of the testing. Linear regression with 95% confidence bands is shown.

**Figure 4 diagnostics-15-01797-f004:**
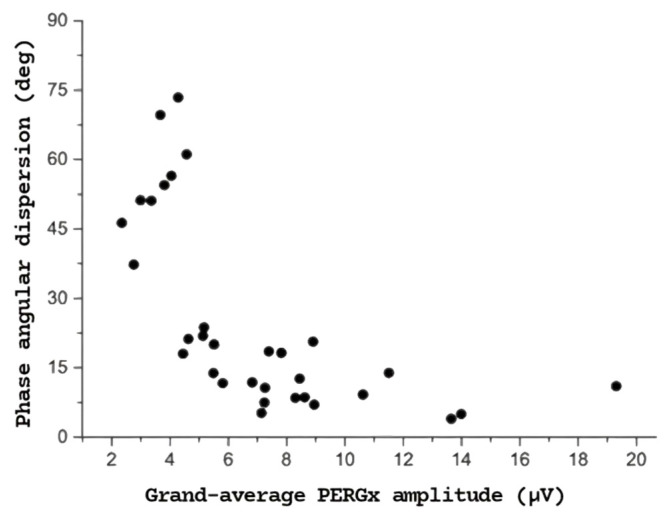
Scatter plot of PERGx phase angular dispersion values as a function of corresponding grand-average PERGx amplitudes across both patient groups taken together, showing a significant (*p* < 0.01) inverse correlation.

**Table 1 diagnostics-15-01797-t001:** Demographic and clinical characteristics of the study population across the three groups. The IOP values reported for the patient groups are intended as treated IOP values. The statistical significance (*p*-value) of comparisons between group parameters is shown. NS indicates non-significant differences.

	Normal Subjects (n = 20)	OHT (n = 20)	OAG (n = 20)	ANOVA *p*
Demographical data				
Age, y: mean ± SD (*range*)	56.35 ± 3.45 *(51*–*66)*	57.35 ± 6.94 *(43*–*66)*	58.8 ± 4.06 *(50*–*65)*	NS
Male, No. (%)	8 (40)	10 (50)	12 (60)	NS
Female, No. (%)	12 (60)	10 (50)	8 (40)	NS
IOP, mmHg: mean ± SD (*range*)	14.8 ± 2.24 *(10*–*18)*	20.8 ± 2.14 *(17*–*24)*	16.55 ± 4.35 *(11*–*24)*	<0.01
Visual field parameters, dB: mean ± SD (*range*)				
MD	0.29 ± 1.33 *(−2.06* to *1.73)*	−0.05 ± 1.35 *(−1.81* to *2.27)*	−5.31 ± 6.4 *(−27.05* to *0.39)*	<0.01
PSD	1.72 ± 0.6 *(1.22*–*4.04)*	1.95 ± 0.68 *(1.45*–*4.63)*	6.18 ± 5.08 *(1.81*–*17.39)*	<0.01

Significant differences were found across the groups for both MD and PSD averages (*p* < 0.01). Post hoc analysis using the Tukey test indicated that both perimetric indices were significantly different (*p* < 0.01) when comparing OAG to both OHT and normal eyes, while no significant differences were observed between OHT and normal eyes.

**Table 2 diagnostics-15-01797-t002:** Functional PERGx and structural (OCT) characteristics of the study population across the three groups. The statistical significance (*p*-value) of comparisons between group parameters is shown.

Variable: mMan ± SD (*Range*)	Normal Subjects	Glaucoma Suspects	Glaucoma Patients	ANOVA *p*
PERG parameters: mean ± SD (*range*)				
Delta amplitude (µV/packet)	−1.23 ± 2.69 *(−7.42* to *4.46)*	0.75 ± 3.49 *(−5.95* to *8.59)*	−1.71 ± 2.15 *(−4.73* to *3.1)*	<0.05
Phase angular dispersion (deg)	9.02 ± 8.14 *(2.18*–*40.55)*	19.31 ± 18.82 *(3.95*–*30.36)*	16.32 ± 7.15 *(7.46*–*37.3)*	<0.05
Grand-average scalar amplitude (µV)	11.88 ± 5.04 *(3.53*–*22.88)*	8.18 ± 4.13 *(3.89*–*19.31)*	7.51 ± 2.43 *(2.76*–*11.51)*	<0.01
Grand-average scalar phase (deg)	−3.96 ± 21.48 *(−45.35* to *26.2)*	15.28 ± 25.68 *(−33.35* to *63.77)*	18.61 ± 17.99 *(−13.85* to *46.07)*	<0.01
OCT parameters, µm: mean ± SD (*range*)				
Average RNFL thickness	91.05 ± 7.68 *(76*–*101)*	86.8 ± 6.76 *(72*–*98)*	68.45 ± 6.24 *(58*–*79)*	<0.01
Average GCIPL thickness	81.15 ± 4.91 *(73*–*90)*	79.1 ± 4.92 *(70*–*85)*	63.9 ± 6.10 *(49*–*71)*	<0.01

The delta of the PERGx scalar amplitude differed significantly among the groups (*p* < 0.05). In particular, patients with OAG had more negative average delta values compared to both OHT patients (*p* < 0.05) and normal subjects (not significant). Comparison between OHT patients and normal subjects was not statistically significant.

## Data Availability

The original contributions presented in this study are included in the article. Further inquiries can be directed to the corresponding author.

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
