# Peer review of "Steady-State PERG Adaptation Reveals Temporal Abnormalities of Retinal Ganglion Cells in Treated Ocular Hypertension and Glaucoma"

_diagnostics, 2025, doi:10.3390/diagnostics15141797_

Round 1
Reviewer 1 Report
Comments and Suggestions for Authors
The authors present a cross-sectional study investigating the adaptive dynamics of steady-state PERG responses in patients with OHT, OAG, and normal subjects. Using a standardized PERGx paradigm, they measured changes in second-harmonic amplitude and phase over time to characterize RGC function and its adaptation. The study reports that both OHT and OAG eyes showed altered phase dynamics and greater phase angular dispersion compared to normal subjects, suggesting temporal abnormalities in RGC responses even in the absence of structural or perimetric glaucomatous damage. The authors propose that PERG phase parameters could provide an early indicator of functional RGC dysfunction and complement existing diagnostic methods.
1. In Figures 2 and 3, the authors show a significant correlation between PERG phase angular dispersion and intraocular pressure (IOP) in the OHT group. However, it would strengthen the manuscript to also report whether similar correlations were tested in the normal control and OAG groups, to clarify whether this relationship is unique to OHT.
Additionally, I am concerned about the interpretation that the OHT group demonstrated higher phase angular dispersion than the OAG group, despite lacking optic disc or visual field abnormalities. Could the authors elaborate on how this apparent temporal dysfunction is mechanistically explained in the absence of structural or perimetric glaucomatous damage? This point deserves more discussion.
2. The manuscript reports that no significant correlations were found between PERG parameters and structural or functional clinical variables (e.g., RNFL thickness, GCIPL thickness, visual field indices). This raises some concern about how PERG parameters can be validated as clinically meaningful glaucoma diagnostics. Could the authors please present the correlation data explicitly (e.g., a table of correlation coefficients and p-values) and further discuss how PERG metrics could still serve as an early biomarker, even if they do not correlate with structural or perimetric measures?
Furthermore, could the authors address whether the observed PERG impairments might be transient or reversible changes associated with IOP modulation, rather than reflecting permanent retinal ganglion cell damage? This clarification would help the reader understand the long-term diagnostic value of PERG.
Author Response
Comments 1: In Figures 2 and 3, the authors show a significant correlation between PERG phase angular dispersion and intraocular pressure (IOP) in the OHT group. However, it would strengthen the manuscript to also report whether similar correlations were tested in the normal control and OAG groups, to clarify whether this relationship is unique to OHT.
Additionally, I am concerned about the interpretation that the OHT group demonstrated higher phase angular dispersion than the OAG group, despite lacking optic disc or visual field abnormalities. Could the authors elaborate on how this apparent temporal dysfunction is mechanistically explained in the absence of structural or perimetric glaucomatous damage? This point deserves more discussion.
Response 1: We tested this correlation in the OAG and normal groups. The correlation was not statistically significant in either group, as now shown in the updated correlation matrix (Supplementary Table S1). This suggests that the inverse correlation between phase dispersion and IOP is specific to the OHT group and may reflect early RGC dysfunction prior to structural or perimetric glaucomatous damage.
We expanded the Discussion to emphasize that the increased phase dispersion may reflect a functional instability or subclinical dysfunction of RGCs in OHT eyes, preceding structural or perimetric damage.
Comments 2: The manuscript reports that no significant correlations were found between PERG parameters and structural or functional clinical variables (e.g., RNFL thickness, GCIPL thickness, visual field indices). This raises some concern about how PERG parameters can be validated as clinically meaningful glaucoma diagnostics. Could the authors please present the correlation data explicitly (e.g., a table of correlation coefficients and p-values) and further discuss how PERG metrics could still serve as an early biomarker, even if they do not correlate with structural or perimetric measures?
Furthermore, could the authors address whether the observed PERG impairments might be transient or reversible changes associated with IOP modulation, rather than reflecting permanent retinal ganglion cell damage? This clarification would help the reader understand the long-term diagnostic value of PERG.
Response 2: A new supplementary table has been added, showing full Pearson correlation matrices between PERG parameters (Amplitude, Phase Angular Dispersion, Delta Amplitude) and clinical structural/functional variables (RNFL, GCIPL, MD, PSD, and IOP), separately for the three groups (Normal subjects, OHT, OAG).
Statistically significant correlations are marked with * (p < 0.05) and ** (p < 0.01).
These data are now provided both in the manuscript (Supplementary Material) and directly below for the reviewer’s convenience.
We clarified in the Discussion that due to the cross-sectional nature of our study, this cannot be directly assessed. However, based on previous findings, we hypothesize that some of these functional alterations could be reversible, particularly in response to IOP modulation or neuroprotective interventions.

Reviewer 2 Report
Comments and Suggestions for Authors
This is a further step in Porciatti's pioneering studies, with an exploration of PERGx temporal abnormality. Lines 357-60 contain an important message: "shed further light on early changes in preserved inner retina of OHT eyes. These changes may precede any detectable manifestation in OHT. This represents a significant leap forward in the staging of patients 'in the glaucoma area,' expanding the diagnostic potential of PERG in this population." Therefore, worldwide Ophthalmologists and Glaucoma Labs must take into account that (line 408) ‘‘phase dispersion, rather than amplitude reduction, may serve as a sensitive early marker of RGC dysfunction in ocular hypertensive or glaucomatous patients.’’
Congratulations on the innovative results that continue to be extracted from electrophysiological research.
Check for Funding statement: to be clarified.
Author Response
Comments: This is a further step in Porciatti's pioneering studies, with an exploration of PERGx temporal abnormality. Lines 357-60 contain an important message: "shed further light on early changes in preserved inner retina of OHT eyes. These changes may precede any detectable manifestation in OHT. This represents a significant leap forward in the staging of patients 'in the glaucoma area,' expanding the diagnostic potential of PERG in this population." Therefore, worldwide Ophthalmologists and Glaucoma Labs must take into account that (line 408) ‘‘phase dispersion, rather than amplitude reduction, may serve as a sensitive early marker of RGC dysfunction in ocular hypertensive or glaucomatous patients.’’
Congratulations on the innovative results that continue to be extracted from electrophysiological research.
Check for Funding statement: to be clarified.
Response: We have revised and updated the reference list to improve relevance and timeliness, reducing self-citations. However, we note that the literature on PERG adaptation remains limited.
We are sincerely grateful for the reviewer’s encouraging remarks regarding the innovative contribution of our study to the field of electrophysiological research.
We confirm that no external funding was received for this work, and this has been explicitly clarified in the revised manuscript under the “Funding” section.
